# Anesthetics drug wastage and preventive strategies: Systematic review

**Meseret Firde Habte** [1]*, **Biresaw Ayen Tegegne**[2], **Tikuneh Yetneberk Alemayehu**[1]

**1** College of Health Sciences, Debre Tabor University, Debre Tabor, Ethiopia, **2** College of Medicine and Health Sciences, University of Gondar, Gondar, Ethiopia

* mesiwyeabeye@gmail.com

**Data Availability Statement:** All relevant data are within the manuscript and its Supporting Information files.

## Abstract

### Background

Surgical Patients and hospitals are now facing financial strain due to direct anesthetic demand as a result of the development of new anesthetic drugs, equipment, and techniques. Up to 15% of a hospital's pharmacy budget is currently allocated to anesthetic drug expenses. Drug wastage during anesthesia practice is a widespread hidden source of healthcare waste that leads to anesthetic drug shortages as well as poor operating room efficiency. On the other hand, despite the fact that it is preventable in the vast majority of cases, it is well described that drug wastage is routinely observed, including in developing countries where the consequences significantly affect both hospitals and patients.

### Methods

This review aims to review the prevalence of anesthetic drug waste across the world and systematically formulate and describe preventive strategies. Relevant publications were identified using systematic searches on databases including Google Scholar, Medline (PubMed), the Cochrane Library, and Embase. In addition, papers were detected and then selected through the Preferred Reporting Items for Systematic Reviews and Meta-Analyses (PRISMA) criteria guidelines and the inclusion and exclusion criteria. Using the predetermined terms and dates from the searching databases, a total of 504 articles were identified. Based on the screening criteria, 16 papers were considered eligible and included in the final review. In addition, the Joanna Briggs Institute (JBI) Manual for Evidence Synthesis was used for evaluating the quality of selected articles. This study is registered on PROSPERO, number CRD42024497044.

### Results

Of the sixteen publications from eleven different nations that were considered suitable for inclusion, only two of them addressed the waste of inhalational anesthetics. In more than half of eligible articles, propofol was the frequently wasted drug that contributed to increased financial loss through drug waste. The first most significant factor contributing to the waste of intravenous and inhalational anesthetics was the disposal of multidrug vials following their use for a single patient and high fresh gas flow, respectively.

**Funding:** The author(s) received no specific funding for this work.

**Competing interests:** The authors have declared that no competing interests exist.

## Conclusion

Anesthetic medication waste is a common occurrence worldwide, despite the fact that it is expensive and has a significant negative impact on operating room efficiency. Because the majority of drug waste is avoidable, preventive measures may lower drug waste and improve patient and hospital efficiency.

## Introduction

Due to the advancement of novel anesthetic drugs, equipment, and procedures, anesthetic costs are becoming a significant financial burden for both patients and hospitals [1]. Particularly, the considerably increased cost of anesthetic medicines has significant consequences, especially in developing countries [2]. According to statistics, operation rooms (OR) account for 40% of total hospital expenditure, yet anesthetic medications account for 5–15% of a hospital's pharmacy budget [1, 3]. Possible changes in the individual responses of patients, unused drugs, and opened ampoules or vials because of sterility concerns make some amount of drug waste unavoidable in anesthesia [2, 4, 5].

Numerous pieces of data indicate this anesthetic medication waste may occur at any time of the perioperative period, yet it can happen most frequently in an emergency [6, 7]. Wastage is also described as being particularly prevalent with intravenous drugs due to the fact that these drugs, as opposed to inhaled anesthetic agents, are packaged in specific amounts in ampoules or vials [4, 8]. These drugs can only be withdrawn after the ampoules are broken or after the rubber stopper is penetrated by the hypodermic needle. Thus, they need to be used within a specific period due to concerns regarding infection control and contamination. Partially used ampoules or syringes or loaded but unused syringes are usually discarded after the end of a procedure [9, 10].

Additionally, it is emphasized that a significant portion of the inhalation agents are not absorbed by the patient but rather are released into the atmosphere through the waste gas scavenging mechanism [10, 11]. When computing medication waste in the operating room, volatile anesthetics, which make up more than 20% of anesthesia expenses, must also be taken into consideration. This is especially important now that newer and more expensive agents, such as sevoflurane, are in clinical use [12, 13]. On the other hand, such wasted drugs have been shown to contribute significantly to the cost of intraoperative anesthesia care and lead to drug shortages. In addition to the hospital's financial costs, discarded medications like propofol contaminate the environment and may have harmful ecological consequences. Moreover, it can increase occupational hazards for health care and sanitary workers and cause environmental pollution [3, 4, 14]. Cost-reduction strategies have gained considerable significance due to rising anesthetic drug prices and health care costs, particularly in developing countries. Inferring that decreasing drug waste is a key area for lowering anesthetic costs without compromising the quality of care delivered [15, 16].

Preventing anesthetic drug waste is critical for increasing OR efficiency, reducing costs, and enhancing patient safety. Scientific evidence underscores the multifaceted benefits of efficient drug management, from economic savings and environmental sustainability to improved operational workflows and patient outcomes. Although potential sources of anesthesia-related waste are not limited to drugs, for the purposes of the current review, we focused on anesthesia-related drug waste. The first reason is that it has been claimed that a hidden source of healthcare waste [5]. Another explanation is that, with the current ongoing increase in drug costs, anesthetic drug wastage directly contributes to anesthetic drug shortages as well as poor

hospital efficiency in surgical patient care [17]. Despite the fact that there are several original studies and a few brief reports on the topic, to the best of our knowledge, this is the first systematic evaluation that examined most recent publications for the waste of anesthetic drugs and their preventative measures taken to decrease them, therefore enhancing the hospital's anesthetic clinical practice in a safe and cost-effective manner.

## Materials and methods

To find the relevant publications, a computerized systematic search of databases, including Google Scholar, Medline (PubMed), Cochrane Library, and Embase, was conducted. Databases were searched using the following terms: anesthesia, drug waste, anesthetic drug, risk factors, predisposing factors, intravenous anesthetics, inhalational anesthetics, operating room efficiency, medication in the operating room, surgery, preventive strategies, and waste in the operating room. Additionally, the aforementioned search phrases were combined with boolean operators (AND, OR) for additional searches.

Papers were detected and then selected through the PRISMA (Preferred Reporting Items for Systematic Reviews and Meta-Analyses) criteria guidelines and the inclusion and exclusion criteria [18]. Fig 1 Only peer-reviewed full original scientific papers, written in English, published after 2010 and which presented the magnitude and/or causes of anesthetic drug waste in OR, were considered in this review (**S1 File**). The publications that were evidently unrelated to the topic (that is, unrelated to the prevalence, significance, or predicting factors of drug wastage in OR) or those that solely addressed other anesthetic wastes (such as waste pertaining to operating room supplies or labor or cost evaluation of only utilized anesthetic drugs) were then excluded from the review. Additionally, publications that solely detailed anesthetic drug waste in settings other than operating rooms (such as intensive care units, emergency rooms, catheterization suites, ambulatory surgery centers, and post-anesthesia care units) were disregarded.

Using the predetermined terms and dates from the searching databases, a total of 504 articles (latest search: December 20, 2023) were identified. Based on the screening criteria, which included duplication and a bias-free methodological aspect, 16 papers were considered to be eligible and included in the final review. In addition, the Joanna Briggs Institute (JBI) Manual for Evidence Synthesis was used for evaluating the quality of selected articles [19]. Table 1 Two investigators used a pre-designed data extraction tool in Microsoft Excel 2016 to extract data independently. Any disputes were settled by arbitration by a third investigator or by consensus.

Data that was extracted included the study's geographic location, design, sample size, usage of one or more recruitment sites, types of medications used in the analysis, and the frequency and contributing factors to drug waste in operating rooms. Furthermore, any statistical results-based preventive strategies for drug waste were also extracted and reported if provided in the manuscript. Additionally, the publications were assessed for age, sex, or type of surgery restrictions, as well as whether the study only involved specific types of anesthetic drugs or a particular surgical urgency. Finally, the main findings, including type, extent, cost, reasons, and recommendations for drug waste during anesthesia practice, were summarized, compared, and discussed systematically. The protocol was registered in the International Prospective Register of Systematic Reviews (CRD42024497044).

## Result

By using the search parameters, 504 papers in total were found. Following the screening process and the application of inclusion and exclusion criteria, 22 of these were deemed

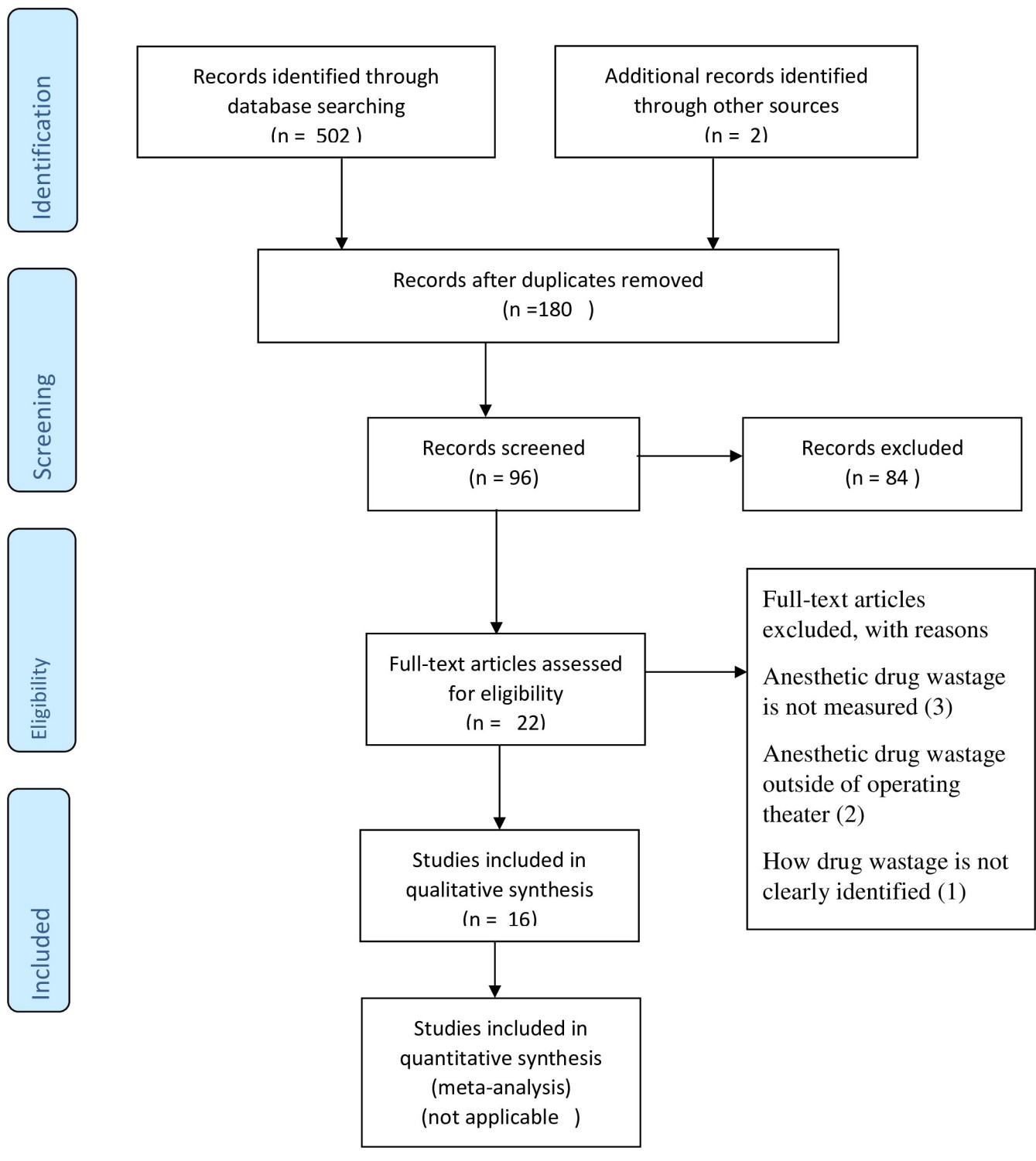

**Fig 1. PRISMA flow diagram of the search strategy.**

appropriate for full-text evaluation, and 16 publications were included in the final review. Table 2 The studies that were analyzed covered the prevalence as well as possible reasons or preventative measures for medication waste during anesthesia practice. The selected articles

**Table 1. Critical appraisal of included studies using the JBI prevalence critical appraisal checklist.**

| | Target population | Sampling | Sample size | Description of participants and setting | Coverage of identified sample | Methods to identify outcome | Reliability in outcome measurement | Appropriate statistical analysis | Response rate | Total |
|---|---|---|---|---|---|---|---|---|---|---|
| K peker, 2020 | Y | Y | Y | Y | Y | Y | Y | Y | Y | 9/9 |
| Amucheazi Adaobi O et al, 2013 | Y | Y | Y | Y | Y | Y | Y | N | Y | 8/9 |
| Kapil Chaudhary et al,2012 | Y | Y | Y | Y | Y | Y | Y | N | Y | 8/9 |
| Dr.Ch.Deepthi Reddy et al,2018 | Y | Y | Y | Y | Y | Y | Y | Y | Y | 9/9 |
| Veena Reshma D'Souza et al,2020 | Y | Y | Y | Y | Y | Y | Y | Y | Y | 9/9 |
| Hannah Dee, 2012 | Y | Y | Y | Y | Y | Y | Y | Y | Y | 9/9 |
| Suvarna Kaniyil et al, 2017 | Y | Y | Y | Y | Y | Y | Y | Y | Y | 9/9 |
| Carrie Leigh Hamby Atcheson et al, 2015 | Y | Y | Y | Y | Y | Y | Y | N | Y | 8/9 |
| Sonali Ramakant More et al,2015 | Y | Y | Y | Y | Y | Y | Y | Y | Y | 9/9 |
| Rohit Malhotra et al, 2020 | Y | Y | Y | Y | Y | Y | Y | N | Y | 8/9 |
| Russell F et al,2012 | Y | Y | N | Y | Y | Y | Y | Y | Y | 8/9 |
| Azfar Athar Ishaqui et al,2023 | Y | Y | N | Y | Y | Y | Y | Y | Y | 8/9 |
| Hailu et al,2017 | Y | Y | Y | Y | Y | Y | Y | Y | Y | 9/9 |
| EZEMA et al,2020 | Y | Y | Y | Y | Y | Y | Y | Y | Y | 9/9 |
| Bala G. Nair et al,2023 | Y | Y | N | Y | Y | Y | Y | Y | Y | 8/9 |
| C. Lejus et al, 2012 | Y | Y | Y | Y | Y | Y | Y | Y | Y | 9/9 |

are from eleven different countries, including Ethiopia and Nigeria. Eleven studies looked at how much medication was used, how much was wasted, and how much money was lost due to avoidable drug waste [2, 8, 11, 20–24]. One of the studies incorporates material waste into its primary outcome, medication waste in the operating room [8]. The majority of studies on intravenous anesthetic drug loss consider drug waste when medications are loaded into syringes but are either partially or completely unused or discarded at the end of the surgical operation [6–8, 11, 25].

## Discussion

Though utilizing cost-effective anesthetic agents and surgical supplies has been shown to improve the quality of treatment while lowering costs, reducing anesthetic drug waste is also critical for improving operating room use, especially in developing countries [29]. The purpose of this review was to investigate the prevalence of anesthetic drug waste in the operating room and identify relevant preventative strategies. The primary finding of the publications included in this systematic review indicates that there is a large waste of anesthetic drugs in the operating room (OR), which has been related to the hospital's deteriorating financial efficiency. It

**Table 2.** Summarizing the key findings of the studies included in this systematic review.

| Author | Title | Study area | Design | Sample size | Result of the study |
|---|---|---|---|---|---|
| K peker, [25] | The Wastage and Economic Effects of Anesthetic Drugs | Chicago | Prospective, observational design | 363 | • The highest total loss cost was rocuronium and propofol.<br>• The cost of lost drugs would decrease if drugs were prepared according to the type of the anaesthesia required after a rapid evaluation when the patient comes to the operating room. |
| Amucheazi Adaobi O et al, [26] | Drug consumption and wastage during anaesthesia | Nigeria | Prospective, observational study | 356 | • Considering the cost, the most wasted drug was propofol followed by 0.5% bupivacaine.<br>• Regular system audit and creating awareness through posters placed in the theatres may help reduce wastage. |
| Kapil Chaudharyet et al, [11] | Anesthetic drug wastage in the operation room | India | Prospective, observational study | 98 | • Lignocaine (5ml), succinylcholine (2ml), and atropine (3ampoules) were loaded in each case.<br>• A total of 350 ml of propofol was loaded in cases performed under regional anesthesia, which was not used at all.<br>• The wastage could have been much less had these vials not been loaded and could have been used in other patients. |
| Ch.Deepthi Reddy et al, [7] | Drug Wastage and Cost Analysis of Anaesthetics | India | Prospective, observational study | 150 | • The drug left in syringe of bupivacaine was 50% in 79 cases, propofol was 1.25% in 2 case among 40 cases<br>• Instead of 4ml ampoules using 2ml ampoules of bupivacaine can decrease wastage and patient economic burden. |
| Veena Reshma et al, [21] | Drug Wastage and its Financial Cost in Anaesthesia | United kingdom | Clinical audit | 60 | • Propofol contributed maximum to the financial loss being 29.27% of the total loss, followed by bupivacaine 18.95%<br>• The reason for wastage in the study was disposal of the multidrug vials (MDV) after using for a single patient. |
| Hannah Dee, [8] | Drug and Material Wastage in Anesthesia Care | Seatle | Prospective, observational study | 164 | • Most wasted drugs, by volume, were propofol, lidocaine, neostigmine, succinylcholine, ephedrine, and phenylephrine. A common source of drug waste in the study was the result of excessive amounts of drugs in vials and syringes.<br>If the contents of an ampoule are likely to be used on more than one patient, practitioners could draw up drugs into several syringes, or "split doses" |
| Suvarna Kaniyil et al, [2] | Financial Implications of Intravenous Anesthetic Drug Wastage in Operation Room | Italy | Prospective, observational study | 644 | • Drugs like 2 ampoules of atropine, 30 mg mephentermine or ephedrine, 10 ml propofol, 100 mg succinylcholine, were routinely loaded for all cases.<br>• The average daily cost of wasted drug was maximum for vecuronium followed by propofol |
| Carrie Leigh etal, [20] | Preventable drug waste among anesthesia providers: | USA | Retrospective observational study | 543 | • The average cost of preventable drug waste for all observed cases was $3.90 per case.<br>• Potential savings with the use of prefilled syringes for some commonly used anesthetic drugs. |
| Sonali Ramakan et al, [27] | Intravenous Anaesthetic Agents | India | Clinical audit | 200 | • The maximum % of waste of loaded drugs was seen with atropine (79%)<br>• Out of 8200mg propofol loaded 3000mg of propofol was wasted<br>• Appropriate measures can effectively decrease the cost in terms of un-utilized drugs |
| Rohit Malhotra etal, [24] | Cost identification analysis of general anesthesia | India | Prospective, observational study | 258 | • The most commonly wasted drugs were muscle relaxants contributed 58% of total wastage.<br>• Instead of full vial, withdrawing drug as required would have decreased wastage. |
| Russell F et al, [4] | Propofol Wastage in Anesthesia | New York | Prospective, observational study | | • Propofol accounted for 45% of the total drug waste<br>• Reducing the size of propofol vials reduced the wastage of propofol |
| Bala G. Nair et al, [12] | Reducing Wastage of Inhalation Anesthetics Using Real-time Decision Support | Seatle | Prospective interventional study | | • Real-time notification for the use of lower fresh gas flows (1-2L/min) is an effective way to reduce inhalation agent wastage |

*(Continued)*

**Table 2.** (Continued)

| Author | Title | Study area | Design | Sample size | Result of the study |
|---|---|---|---|---|---|
| Azfar Athar Ishaqui et al, [23] | Intravenous narcotics and controlled drugs wastage and their financial impact | Saudi Arabia | Prospective, observational study | | • The highest ampoule wastage was observed for Morphine 10 mg formulations (1956 ampoules). <br>• Shifting to prefilled syringes supplied by pharmacies, could result in significant savings. |
| Hailu et al, [6] | Wastage of Anaesthetic and Analgesic Agents | Ethiopia | Clinical audit | 86 | • A total of 1967.8 Ethiopian birr (89.44 USD) were the cost of wasted drugs, of these bupivacaine was maximum (33.8%). |
| EZEMA et al, [22] | Anaesthetic Agent Usage and Wastage during Caesarean Deliveries | Nigeria | Prospective, observational study | 117 | • The wastage was found more during the use of hyperbaric bupivacaine formed $210.10 of the total cost of wasted local anaesthetic agents <br>• Practice guidelines, feedback to provider, and monitoring of waste reduction for drug usage may immensely yield positive outlook. |
| C. Lejus et al [28] | Atropine and ephedrine: a significant waste in the operating theatre | France | Retrospective, observational study | 27 705 | • The rate of wastage differed between units with extremes of 39% and 67% for ephedrine and 69% and 91% for atropine <br>• Preoperative systematic preparation of atropine and ephedrine is rarely justified |

also negatively affects the environment because improper disposal methods can contaminate water supplies and possibly cause health risks to humans [11].

Propofol is an expensive drug with several advantages, including faster recovery, reduced postoperative nausea and vomiting, and a vomiting, and a shortened stay in the post-anesthesia care unit (PACU) [30]. Its waste has greater environmental implications as it does not degrade in nature, accumulates in body fat, and is toxic to aquatic life [4]. Despite these benefits and its potential environmental impact upon disposal, according to the results of numerous studies, propofol was shown to be the most wasted medicine, accounting for 45% of the total drug waste measured in milliliters [4, 31]. Due to being partially unused after loading into a syringe and multidrug vial (MDV) disposal following a single patient's use, propofol was determined to be the most frequently wasted drug in more than half of the reviewed articles [4, 21, 24].

An observational study of 356 procedures conducted by Amucheazi Adaobi O et al. revealed that propofol was one of the most commonly wasted drugs, accounting for more than 50% of the overall waste cost [26]. Similarly, an Indian study that examined the amount and financial implications of drugs left unutilized in syringes or vials after completion of each case found that the cost of wasted anesthetic drugs for the period of the study accounted for 46.57% of the entire cost of drugs loaded (Rs. 16,044.01). Where the cost of propofol waste contributed the most to overall waste, accounting for 56.27% [11].

In terms of analgesics, the findings of three studies revealed a significant amount of analgesic drug wastage, including narcotics, during anesthesia practice. A study by Hailu et al. from Ethiopia discovered that tramadol, diclofenac, and fentanyl were among the most loaded in syringes but not used, resulting in more than 273 Ethiopian birr waste during the two-week study period [6]. Morphine and pethidine were the most lost drugs due to not being utilized after loading into the syringe, accounting for 26.33% and 23.33% of total intravenous anesthetic drug wastage, respectively, according to a related study from Nigeria [12]. The analysis of annual narcotic wastage in Saudi Arabia revealed that the highest ampoule wastage was observed for Morphine 10 mg formulations (1956 ampoules), and the authors emphasized that shifting to prefilled syringes with calculated doses for a specific patient provided by pharmacies could result in significant waste reduction and cost savings [6].

Despite the fact that wasted inhalational anesthetic gases are expensive and may negatively impact the environment, there have been limited studies on the subject. Likewise, only two of the reviewed papers considered inhalational agent waste [12, 24]. This could be because inhalational medicines are delivered in liquid formulations but are administered as vapors; as a result, it is more challenging to determine how much inhalation agent is wasted [32–34]. For a prospective study of 258 adult patients, the authors purposely divided the maintenance phase of anesthesia into low (FGF ≤2 L/min) and high (FGF >2 L/min) sevoflurane and isoflurane with a targeted MAC of 0.8–1.2 [24]. The result of the same study showed that the mean cost in high flow groups was higher than that in low flow groups and concluded that low flow anesthesia with isoflurane is more cost-effective as compared to high flow techniques [12].

Moreover, significant drug wastage has also been documented following the use of resuscitation medications (atropine, glycopylorate, adrenaline, epinephrine, phenylephrine, mephentermine), local anesthetics (bupivacaine, lignocaine), neuromuscular medications (atracurium, pancuronium, rocuronium, and vecuronium), and other medications (neostigmine, diazepam, and midazolam) [6, 11, 25, 27].

## Preventive strategies for drug wastage in operating theaters

Waste-reduction measures in healthcare are becoming more and more pertinent and deserving of consideration in an era where hospital and patient costs are rising [20, 35]. Even if drug use should not and cannot be restricted to save expenses and so compromise patient care, the easy procedures recommended would be advantageous in reducing drug waste [2, 20, 36]. As a result, the following preventive methods are discussed after a thorough review of all relevant studies published after 2010.

## Split dose

If the contents of an ampoule are likely to be used on more than one patient, practitioners could draw up drugs into several syringes, or "split doses" [8]. To ensure an efficient distribution of drugs, based on body weight and according to the need and demand of each case, prefilled syringes will be provided. And any remaining amount in multidose vials can be used to provide prefilled syringes for additional cases [23]. However, it is not necessarily extended to the preparation of other drugs, such as propofol, that may not be used during the case but are discarded after the procedure [4]. On the other hand, when the practice of split dose is considered, special attention should be given to avoid excessive volumes of medications in syringes, which is one of the frequent sources of drug waste that multiple studies have highlighted [6, 22].

In resource-constrained contexts, the pressure to save anesthetic medicines is greater, making vial sharing more likely. On the other hand, scientific data challenges this approach by emphasizing significant associated hazards, particularly cross-contamination between patients and infection control [37–39]. Furthermore, it has been discovered that splitting doses from a single vial might result in dosing inaccuracies due to variations in the volume drawn into each syringe, which can be particularly problematic with anesthetics that require precise dosing [40, 41]. As a result, while sharing vials can help minimize drug waste and associated expenses, the benefits must be balanced against the potential risks of healthcare-associated infections and pharmaceutical safety [39]. Furthermore, adherence to regulatory authorities' rigorous restrictions prohibiting the practice of sharing vials among patients, as well as ongoing education and training of healthcare personnel about the risks involved with vial sharing, are critical to ensuring patient safety and institutional credibility [42].

## Propofol

Disposal of the multidrug vials (MDV) after using them for a single patient was the documented primary cause of propofol waste. Similarly, findings have revealed that opening 50-ml vials specifically for induction and filling 50-ml syringe pumps for the maintenance of a short case may be the principal causes of this wastage. Additionally, because it is advised to discard vials after 6 hours of opening, opening 50 ml vials on days with a short OR case list, one major case, or cases under regional anesthesia would result in a significant amount of drug left in the vial or syringe being wasted [4, 20, 26].

Given the high cost, environmental impact, and short shelf life of propofol in a syringe, it is important to optimize its use, ideally by preparing an appropriate dose when only it is actually needed. Results indicated that no patient needed more than 20 ml of propofol to get induction of anesthesia. Consequently, in order to prevent medication waste from leftover syringes, it would be recommended to load up to the highest limit range for induction (2.5 mg/kg), which would be less than 20 ml for a patient weighing an average of 60–70 kg [20].

Additionally, purchasing 20-ml vials would be another better option for propofol-related waste management. Even though smaller vials have a higher unit price than 50-ml vials, it may be possible to reduce actual financial loss by lowering overall waste. As a result, selecting the appropriate vial and communicating about the anesthetic plan in advance may help reduce waste. Additionally, having a vial of propofol on hand in the operating room is also helpful to prevent waste as a result of putting a higher volume of propofol into syringes for cases including regional anesthesia and then leaving them unused [4, 43].

## Inhalational anesthetic agents (IAA)

It is well known that there is potential for cost reduction in the field of anesthesia by preventing the wastage of inhalation agents, as they account for a significant portion (20–25%) of the costs associated with anesthetic drugs [3, 10]. Scientific reports indicate a clear relationship between the fresh gas flow rate and the amount of inhaled anesthetic agent wastage [44]. A high flow rate reduces the amount of exhaled anesthetic gas that is rebreathed and increases the amount of vaporized inhaled anesthetic agent. Hence, if the amount of anesthetic gas vaporized exceeds what is partitioned from the gas phase into the lung and brain tissues, excess anesthetic gas ends up being vented into the atmosphere through the waste gas scavenging system [12, 44].

Using a low fresh gas flow rate, on the other hand, maximizes the rebreathing of exhaled anesthetic gas and reduces the amount of anesthetic gas wasted by venting into the atmosphere. The efficiency of desflurane over a 2-hour anesthetic period with 4.4 l/min vs. 1.0 l/min fresh gas flows was demonstrated to be 0.07 and 0.23, respectively, in the study. This indicates that the patient absorbs just 7% of the desflurane, with the remaining 93% going to waste at a flow rate of 4.4 l/min [44].

To reduce wastage of excess agents, a study was proposed to evaluate the effectiveness of real-time notification by the Smart Anesthesia Messenger system that reminds the anesthesia provider to reduce fresh gas flow within the limits recommended by the Food and Drug Administration of the country. The study was particularly conducted on one baseline and three intervention phases of a surgical procedure under anesthesia using either sevoflurane, isoflurane, or disflorane. The authors reported that they saved 9.5 l of sevoflurane, 6.0 l of desflurane, and 0.8 l of isoflurane per month, translating to an annual savings of $104,916. They also concluded that real-time notification is an effective way to reduce inhalation agent usage by decreasing excess fresh gas flow [12].

## Neuromuscular-blocking drugs (NMBD)

According to Kaniyil (2017), vecuronium and rocuronium, in particular, were discovered to be the second most often wasted medications, accounting for 35.21% of the drug waste in Oregon [20, 27]. This can be reduced by loading the medication appropriately for each case (based on the kg of body weight), allowing the sterile medication that remains in the multidose vial to be used in subsequent cases rather than discarded [11]. In addition, to reduce waste, it is crucial to make thoughtful decisions about the neuromuscular blocker to use in each situation, including continuously monitoring if a neuromuscular blocker is necessary intraoperatively and loading the medication in accordance with the requirements of the patient [2].

The medication doses for each case will be calculated based on the patient's weight and the typical dose range of that drug for the procedure, and the dose will be shown on the machine prior to the commencement of each case [45]. It would help in determining the patient's dosage requirements and ensuring proper loading, hence reducing waste.

## Local anesthetics

Numerous prospective studies of drug waste and cost analyses found significant waste when using a 4 ml vial of bupivacaine for a single patient [7]. Furthermore, the majority of waste related to the loading of 5 ml of lignocaine in each case, mainly for the prevention of pain from propofol injection, is reported. As a result, using 2 ml instead of 4 ml of bupivacaine ampules could reduce the percentage of medication residue in vial or syringe waste and ease the financial burden on patients and hospitals [11]. It is also advised to mix 1–2 ml of lignocaine with propofol or to load it in the syringe and administer it before injecting propofol, rather than loading 5 ml in each case [7]. Additionally, ensuring that lignocaine multidose vials are easily accessible on or next to the anesthetic machine would significantly reduce waste [4, 45, 46].

## Emergency (resuscitation) drugs

In an emergency scenario, preparing the drugs in a syringe can delay treatment and increase dilution error. In many operating settings, it is standard practice to prepare the drugs before induction of anesthesia. As a result, anesthesia care providers commonly draw up one or more syringes of resuscitation drugs, including ephedrine, phenyleprine, and atropine, prophylactically [20]. These emergency drugs are not always needed, and because of concerns about infection control, drugs in syringes incompletely used for one patient are usually not administered to subsequent patients. So that the prepared and left-in-syringe doses may be discarded at the end of a clinical workday [20, 22].

A retrospective study from France was conducted to determine the number of patients given atropine and ephedrine. On 27 705 operations, it was shown that the rate of wastage of atropine and ephedrine differed between units, up to 67% and up to 91%, respectively [28]. Similarly, a related study also revealed that high-acuity drugs, like epinephrine and atropine, are almost always prepared but rarely used, and as a result, almost always wasted [45]. In settings where the anesthesiologist works alone or where anesthesia assistants are not well trained, the practice of loading one ampule of atropine, adrenaline, ephedrine, or mephentermine before surgery may seem relevant in anticipated cases [11, 20, 45]. It may also be possible to predict if resuscitation will be required by taking into account the patient's vascular condition, the extent of sensory block, and the adequacy of fluid preload prior to neuraxial anesthesia.

Additionally, it is also possible to make sure that it is easily accessible intraoperatively in case it is needed to load medication just when necessary. Otherwise, routine preoperative preparation of emergency medications is rarely justified, and a safer approach would be to store

emergency drug ampoules with a saline-filled syringe on or close to the anesthetic machine so that they are ready to load and use when needed. On the other hand, by loading 15 mg of mephentermine sulfate in situations where hypotension is anticipated rather than 30 mg in every case, mephentermine sulfate waste can be reduced by almost half [45].

Furthermore, establishing an emergency drug tray attached to the workstation, systematically arranging and labeling it, and ensuring that it is immediately available on or near the anesthesia machine is a very significant strategy to prevent unwanted loading of all drugs and ensure availability in case of emergency [27].

### Improve provider's awareness of anesthetic drug waste

Every healthcare provider must consider the financial effects of the decisions they make on a daily basis and optimize practice patterns [2]. Despite this growing body of knowledge, lowering drug costs may not be a high priority for the average anesthesia practitioner, and cost awareness remains low. Reducing drug waste may require interventional education in the form of lectures that teach the aforementioned recommended practices, as well as the placement of posters in operating rooms, anesthesia induction rooms, and recovery rooms [22].

Aside from posters, the drug prices on the anesthesia machine are another useful visual reminder of financial constraints. Individual hospital efforts to raise awareness about reducing drug waste and using expensive medications responsibly can be effective. Furthermore, education programs, such as seminars on anesthetic drug waste and drug waste strategies, are recommended [11].

### Audit

Finally, a regular review and audit for the storage, use, and ease of rapid access when necessary of anesthetic drugs should be carried out in order to better understand the efficacy of implementing recommendations and determine whether additional adjustments are needed for a greater reduction in waste [11, 20]. Incentives for practitioners and OR teams who reduced waste would also be a better way to enforce waste-reducing practices and promote competition. The development of forums and regular discussions that focus on reducing the costs related to the waste of anesthetic medication are all very important. Furthermore, there should be a properly functioning incident reporting system that enables the documentation, evaluation, and resolution of challenges related to anesthetic drug wastage [6, 21].

## Conclusion

Anesthetic drug waste is a global issue with high costs and negative effects on efficiency. Reducing waste can improve patient care and hospital operations. Strategies include precise dosing, staff education, and protocol standardization, ensuring safe and effective reduction. On the other hand, while implementing strategies to reduce anesthetic drug waste is crucial for improving operating room efficiency and patient care, it is important to consider some issues, including the risks to infection control and medication stability when sharing vials between patients.

### Limitation

Even though this systematic review includes recent publications from all over the world that may have represented current trends of anesthetic medication waste in operating rooms, the number of eligible articles incorporated has been limited, particularly when it comes to inhalational agents. It makes addressing all the particular issues associated with the avoidance of

anesthetic drug waste challenging. Furthermore, there is little evidence to support certain studies' recommendations for methods of minimizing anesthesia drug waste, such as administering "split doses" when an ampoule's contents are anticipated to be used on several patients. To identify best practices in this regard, we propose conducting long-term primary studies to evaluate the impact of different approaches on waste reduction. Likewise, waste of other important factors that could affect OR efficiency, including personnel expenses, surgical tools, and OR time, is not addressed. Moreover, preventive measures are broader in nature and do not take particular patient characteristics or surgical techniques into account. Thus, it is advised that the identified gaps in this review be included in future research investigations.

## Supporting information

**S1 File.**
(PDF)

## Author Contributions

**Conceptualization:** Meseret Firde Habte.

**Data curation:** Meseret Firde Habte.

**Formal analysis:** Tikuneh Yetneberk Alemayehu.

**Methodology:** Tikuneh Yetneberk Alemayehu.

**Supervision:** Tikuneh Yetneberk Alemayehu.

**Validation:** Biresaw Ayen Tegegne, Tikuneh Yetneberk Alemayehu.

**Writing – original draft:** Biresaw Ayen Tegegne.

**Writing – review & editing:** Meseret Firde Habte.

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
