## [Decision Letter · Decision Letter 0]

10 Mar 2024

PONE-D-24-00988Anesthetics drug wastage and preventive strategies: systematic reviewPLOS ONE

Dear Dr. Habte,

Thank you for submitting your manuscript to PLOS ONE. After careful consideration, we feel that it has merit but does not fully meet PLOS ONE’s publication criteria as it currently stands. Therefore, we invite you to submit a revised version of the manuscript that addresses the points raised during the review process.

**Thank you for submitting your review on this important topic of wastage and financial burden of anesthesia medications. While ****anesthesia wastage is not limited to medications it forms a sizable portion of the economic burden.****I laud the author for choosing a topic of significance even though it has been discussed and studied extensively before. ****The author has enclosed a review of practices and possible causes of drug wastage across the world. ****While the review is extensive some concerns include:****The manuscript requires thorough grammatical revision as the language currently has many errors.****A more detailed environmental impact of inhalation agents and narcotics waste  if included in the studies  could be mentioned.****A mention of billing challenges in multi dose vials and sterility concerns for drugs like propofol would be pertinent.****Some concerns raised by reviewers are attached.**==============================

We look forward to receiving your revised manuscript.

Kind regards,

Shweta Rahul Yemul Golhar, MD

Academic Editor

PLOS ONE

2. PLOS requires an ORCID iD for the corresponding author in Editorial Manager on papers submitted after December 6th, 2016. Please ensure that you have an ORCID iD and that it is validated in Editorial Manager. To do this, go to ‘Update my Information’ (in the upper left-hand corner of the main menu), and click on the Fetch/Validate link next to the ORCID field. This will take you to the ORCID site and allow you to create a new iD or authenticate a pre-existing iD in Editorial Manager. Please see the following video for instructions on linking an ORCID iD to your Editorial Manager account: https://www.youtube.com/watch?v=_xcclfuvtxQ.

Reviewers' comments:

Reviewer's Responses to Questions

**Comments to the Author**

1. Is the manuscript technically sound, and do the data support the conclusions?

Reviewer #1: Yes

Reviewer #2: Yes

2. Has the statistical analysis been performed appropriately and rigorously? 

Reviewer #1: Yes

Reviewer #2: Yes

3. Have the authors made all data underlying the findings in their manuscript fully available?

Reviewer #1: Yes

Reviewer #2: Yes

4. Is the manuscript presented in an intelligible fashion and written in standard English?

Reviewer #1: Yes

Reviewer #2: Yes

5. Review Comments to the Author

Reviewer #1: Well done manuscript on a topic that is critical especially in developing countries. I appreciate their critical appraisal of all papers. I do request they double check their grammar and typos as there are several throughout the paper.

Reviewer #2: This is a systematic review paper summarized the anesthetics drug wastage in the hospitals based on the public data bases, including google scholar, PubMed and Cochrane library and embase. The authors were able to identify 504 articles related to the topic and found only 16 publications can be selected and used for further analysis after going through the PRISMA. Finally, the author concluded that anesthetic medication waste is a common occurrence worldwide, especially propofol, and suggests some preventive measures to lower drug waste. Overall, although lacking potential novelty, the paper is sound and easy to read, the topic is interesting. However, there are some minors should be addressed before publication.

(1) In the title, the authors proposed the preventive strategies to avoid anesthetic drug wastage, like split dose, is there any limitations using this approach? Like splitting dose could also cause wastage?

(2) Propofol and inhalational anesthetics were suggested to be the most waste drugs in OR. However, is there any difference between inhalational drugs? Like isoflurane or sevoflurane?

6. PLOS authors have the option to publish the peer review history of their article (what does this mean?). If published, this will include your full peer review and any attached files.

Reviewer #1: No

Reviewer #2: No

---

## [Author Response · Author response to Decision Letter 0]

21 Mar 2024

Reviewers' Comments to the Authors:

Reviewer 1

Well done manuscript on a topic that is critical especially in developing countries. I appreciate their critical appraisal of all papers. 

Author response: Thank you!

1. I do request they double check their grammar and types as there are several throughout the paper.

Author response: we appreciate you bringing this to our attention. The reviewer is correct, and we have gone over the entire article again and fixed any grammatical issues.

Reviewer 2

Overall, although lacking potential novelty, the paper is sound and easy to read, the topic is interesting. 

Author response: Thank you!

However, there are some minors should be addressed before publication.

1. In the title, the authors proposed the preventive strategies to avoid anesthetic drug wastage, like split dose, is there any limitations using this approach? Like splitting dose could also cause wastage?

Author response: As suggested by the reviewer, we incorporated the general limitations of implementing split doses in our response, which we greatly appreciate. 

2. Propofol and inhalational anesthetics were suggested to be the most waste drugs in OR. However, is there any difference between inhalational drugs? Like isoflurane or sevoflurane?

Author response: Thank you! But even we conducted an unrestricted search to determine whether the incidence of drug waste varies among inhalational agents for the purposes of this review, but we were unable to uncover any information on the subject.

---

## [Decision Letter · Decision Letter 1]

10 Jun 2024

PONE-D-24-00988R1Anesthetics drug wastage and preventive strategies: systematic reviewPLOS ONE

Dear Dr. Habte,

Thank you for submitting your manuscript to PLOS ONE. After careful consideration, we feel that it has merit but does not fully meet PLOS ONE’s publication criteria as it currently stands. Therefore, we invite you to submit a revised version of the manuscript that addresses the points raised during the review process.

 After obtaining further recommendations, I encourage you to address the comments raised by Reviewer 4. A concern raised previously by myself as well is the practice of multiple use vials and possibility of contamination and billing issues. A more balanced approach would be to suggest practice changes that would not raise concerns raised by reviewer 4 and myself. Since there is not much evidence available to support these practices it can be included as limitations of the study. 

We look forward to receiving your revised manuscript.

Kind regards,

Shweta Rahul Yemul Golhar, MD

Academic Editor

PLOS ONE

Reviewers' comments:

Reviewer's Responses to Questions

**Comments to the Author**

1. If the authors have adequately addressed your comments raised in a previous round of review and you feel that this manuscript is now acceptable for publication, you may indicate that here to bypass the “Comments to the Author” section, enter your conflict of interest statement in the “Confidential to Editor” section, and submit your "Accept" recommendation.

Reviewer #2: All comments have been addressed

Reviewer #3: (No Response)

Reviewer #4: All comments have been addressed

Reviewer #5: (No Response)

2. Is the manuscript technically sound, and do the data support the conclusions?

Reviewer #2: Yes

Reviewer #3: Yes

Reviewer #4: Yes

Reviewer #5: No

3. Has the statistical analysis been performed appropriately and rigorously? 

Reviewer #2: Yes

Reviewer #3: Yes

Reviewer #4: N/A

Reviewer #5: I Don't Know

4. Have the authors made all data underlying the findings in their manuscript fully available?

Reviewer #2: Yes

Reviewer #3: Yes

Reviewer #4: Yes

Reviewer #5: Yes

5. Is the manuscript presented in an intelligible fashion and written in standard English?

Reviewer #2: Yes

Reviewer #3: Yes

Reviewer #4: Yes

Reviewer #5: Yes

6. Review Comments to the Author

Reviewer #2: All the comments have been answered by the author and i am satisfied with the author responses. The manuscript is improved by revision.

Reviewer #3: Anesthetic drug wastage may not be the major, or even a minor contributor to drug shortages. Other items such as financial considerations, politics, patents, and even religion may predominate.

In some countries the environmental impact of desflurane and nitrous oxide have removed them from the market.....should more countries adopt that strategy?

Users' education is probably the most important consideration, although anesthesia care providers may be non-plussed by the enormous waste of suture material by surgeons, which far exceeds the $3.90/case quoted by Atcheson. Perhaps the authors might like to highlight anesthesiologists as torch bearers for improved OR efficiency (40% of total hospital expenditure)....less drapes, decreased discarding of full or almost full IV bags, large numbers of surgical instruments and solutions that are not used etc.

The analyses represent 16 studies from 11 countries, including Europe, India, Africa (mostly single studies) and 2 from the US. I see this as a little restricted to label it worldwide...rather a selection of countries.

In some countries like the US, rules, dictated by malpractice suits often, mandate single use vials and discarding of open ampoules or solutions after a certain time even though science has not borne out contamination issues. Reviewing these data and reintroducing them to hospital policy and procedures may also help to revise some current situations (especially before drawn up sugammadex is mandated)

Reviewer #4: In this systematic review, the authors performed a systematic review of the literature to define the financial implication of the anesthetic drugs wastage, and suggest strategies to reduce drug wastage.

Comments:

In my opinion, there are a few points that could be addressed.

1. Although I do appreciate the systematic review performed by the authors, there isn't a lot of literature on the topic. I am not convinced the existing literature is actually helping the authors address the topic other than by highlighting the gaps in the literature.

2. Drug wastage and shortage are important topics, even Moore so since the COVID-19 pandemic. Reductinbg drug wastage is an important point to help reduce drug shortage. The idea of monitoring drug usage and wastage at the institution level, and implementing surveys to monitor it. One aspect that we cannot comprise is safety. Sharing vials between patients etc. should not be promoted as it increases the risk of adverse events.

3. Financial aspects are important, but it is an event more complex topic. We could argue that cost conscious care should be promoted. For example, in many countries sugammadex has become the NMB reversal of choice despite the massive cost of the drug. The same is true for the routine use of dexmedetomidine, fibrinogen concentrate, etc. One should promote cost conscious care in order to reduce cost associated with the use of expansive drugs while cheaper and comparable drugs are available. Is there any data available out there in term of cost-effectiveness for anesthetic drugs? I would suggest the authors to address that.

4. I don't know that the conclusion is truly supported by the results of the systematic review. It is true, but there is an important gap in the literature.

5. As one of my colleagues would argue, the cost of anesthetic drugs only accounts for a small proportion of the overall cost when taking into account surgery, surgical tools, operating room time, ICU costs, etc.

6. I don't think it's been published, but several institutions have saved millions by monitoring drug usage and wastage.

7. The discussion is interesting but they are a lot of other aspects not discussed here.

8. It would be good if the authors could add a central figure or infographic summarizing the topic, their findings, and their suggestions.

Reviewer #5: I do not think that this manuscript meets the standards of Plos One journal. Although I am only reviewing the revised version of this manuscript, but I do not think this is a unique piece of Science. In my opinion, this article is not suitable for publication in Plos. It might be transferred to another journal related to anesthesia/clinical work.

7. PLOS authors have the option to publish the peer review history of their article (what does this mean?). If published, this will include your full peer review and any attached files.

Reviewer #2: No

Reviewer #3: **Yes: **Elizabeth Frost

Reviewer #4: **Yes: **DAVID FARAONI, MD, PhD

Reviewer #5: No

---

## [Author Response · Author response to Decision Letter 1]

23 Jun 2024

Editor’s comment

After obtaining further recommendations, I encourage you to address the comments raised by Reviewer 4. 

A concern raised previously by myself as well is the practice of multiple-use vials and the possibility of contamination and billing issues. A more balanced approach would be to suggest practice changes that would not raise concerns raised by reviewer 4 and myself. Since there is not much evidence available to support these practices, they can be included as limitations of the study.

Author response: Thank you, and we have thoroughly addressed this issue in both the subsection on 'split dose' and in the limitations of the review.

Reviewers' Comments to the Authors:

Reviewer 4

This is a potentially interesting study suitable for the journal's mission and readership.

Author response: Thank you!

1. Although I do appreciate the systematic review performed by the authors, there isn't a lot of literature on the topic. I am not convinced the existing literature is actually helping the authors address the topic other than by highlighting the gaps in the literature.

Author response: We appreciate your valuable insight and have noted the gaps in the manuscript's limitations section.

2. Drug wastage and shortage are important topics, even more so since the COVID-19 pandemic. Reducing drug waste is an important step to help reduce drug shortages. The idea of monitoring drug usage and waste at the institution level and implementing surveys to monitor it. One aspect that we cannot control is safety. Sharing vials between patients, etc., should not be promoted, as it increases the risk of adverse events.

Author response: Again, we want to thank you for your contributions. We have addressed what was suggested and made the necessary adjustments by going over the supporting data in the manuscript's "split dose" and limitation sections.

3. Financial aspects are important, but it is a more complex topic. We could argue that cost-conscious care should be promoted. For example, in many countries, sugammadex has become the NMB reversal of choice despite the massive cost of the drug. The same is true for the routine use of dexmedetomidine, fibrinogen concentrate, etc. One should promote cost-conscious care in order to reduce the cost associated with the use of expansive drugs while cheaper and comparable drugs are available. Is there any data available out there in terms of cost-effectiveness for anesthetic drugs? I would suggest the authors address that.

Author response: We appreciate your input but respectfully decline to include its details in the manuscript for the following reasons:

1. Our review is specifically focused on identifying the prevalence and potential preventive strategies for anesthetic drug waste. Discussing other issues related to operating room efficiency, including cost-conscious anesthesia care, would be outside the scope and title of our review.

1. The publications included in our review do not consider cost-conscious anesthesia care as a preventive strategy for drug waste or as a method for increasing operating room efficiency.

However, we acknowledged the importance of cost-conscious anesthesia care at the beginning of the discussion and have highlighted.

4. I don't know if the conclusion is truly supported by the results of the systematic review. It is true, but there is an important gap in the literature.

Author response: Thank you once more for your contributions. We have rewritten the conclusion to integrate the gaps that readers should consider when applying preventive strategies.

5. As one of my colleagues would argue, the cost of anesthetic drugs only accounts for a small proportion of the overall cost when taking into account surgery, surgical tools, operating room time, ICU costs, etc.

Author response: We completely embraced your suggestion, which is why we have included these issues in the limitations section of the review.

6. I don't think it's been published, but several institutions have saved millions by monitoring drug usage and waste.

Author response: That is accurate, and we have noted that while there are numerous primary studies and non-systematic reviews on the topic, ours represents the first systematic review.

7. The discussion is interesting, but there are a lot of other aspects not discussed here.

Author response: We acknowledge the significance of your feedback, and as per your suggestions, we have integrated considerations such as scientific discussions on vial sharing and an emphasis on cost-conscious care.

8. It would be good if the authors could add a central figure or infographic summarizing the topic, their findings, and their suggestions.

Author response: Dear reviewer, We appreciate your feedback and recognize its importance. However, we believe that Table 2 adequately addresses this question by summarizing the topics covered in each study, along with their findings and suggestions. Introducing an additional table with related information may be redundant. Nevertheless, if you believe it is necessary to create another table to succinctly summarize the key points raised, we are prepared to do so.

---

## [Editor Report · Decision Letter 2]

26 Jun 2024

Anesthetics drug wastage and preventive strategies: systematic review

PONE-D-24-00988R2

Dear Dr. Meseret Firde Habte,

We’re pleased to inform you that your manuscript has been judged scientifically suitable for publication and will be formally accepted for publication once it meets all outstanding technical requirements.

Kind regards,

Shweta Rahul Yemul Golhar, MD

Academic Editor

PLOS ONE
---

## [Editor Report · Acceptance letter]

8 Jul 2024

PONE-D-24-00988R2 

PLOS ONE

Dear Dr. Habte, 

I'm pleased to inform you that your manuscript has been deemed suitable for publication in PLOS ONE. Congratulations! Your manuscript is now being handed over to our production team.

Kind regards, 

on behalf of

Dr. Shweta Rahul Yemul Golhar 

Academic Editor

PLOS ONE